# Intersectionality: Social Marginalisation and Self-Reported Health Status in Young People

**DOI:** 10.3390/ijerph17218104

**Published:** 2020-11-03

**Authors:** Fiona Robards, Melissa Kang, Georgina Luscombe, Catherine Hawke, Lena Sanci, Katharine Steinbeck, Karen Zwi, Susan Towns, Tim Usherwood

**Affiliations:** 1Department of General Practice, Westmead Clinical School, The University of Sydney, Westmead 2145, Australia; melissa.kang@uts.edu.au (M.K.); tim.usherwood@sydney.edu.au (T.U.); 2Australian Centre for Public and Population Health Research, University of Technology Sydney, Ultimo 2007, Australia; 3School of Rural Health, The University of Sydney, Orange 2800, Australia; georgina.luscombe@sydney.edu.au (G.L.); catherine.hawke@sydney.edu.au (C.H.); 4Department of General Practice, University of Melbourne, Carlton 2218, Australia; l.sanci@unimelb.edu.au; 5Discipline of Child and Adolescent Health, University of Sydney, Clinical School at The Children’s Hospital at Westmead NSW, Westmead 2145, Australia; kate.steinbeck@health.nsw.gov.au (K.S.); susan.towns@health.nsw.gov.au (S.T.); 6School of Women’s and Children’s Health, The University of New South Wales, Sydney 2052, Australia; karen.zwi@health.nsw.gov.au; 7Department of Adolescent Medicine, Sydney Children’s Hospital Network, Westmead 2145, Australia; 8The George Institute for Global Health, University of New South Wales, Camperdown 2050, Australia

**Keywords:** young people, adolescents, access to health care, marginalised youth, intersectionalities

## Abstract

Background: The aim of this study was to measure young people’s health status and explore associations between health status and belonging to one or more socio-culturally marginalised group. Methods: part of the Access 3 project, this cross-sectional survey of young people aged 12–24 years living in New South Wales, Australia, oversampled young people from one or more of the following groups: Aboriginal and or Torres Strait Islander; living in rural and remote areas; homeless; refugee; and/or, sexuality and/or gender diverse. This paper reports on findings pertaining to health status, presence of chronic health conditions, psychological distress, and wellbeing measures. Results: 1416 participants completed the survey; 897 (63.3%) belonged to at least one marginalised group; 574 (40.5%) to one, 281 (19.8%) to two and 42 (3.0%) to three or four groups. Belonging to more marginalised groups was significantly associated with having more chronic health conditions (*p* = 0.001), a greater likelihood of high psychological distress (*p* = 0.001) and of illness or injury related absence from school or work (*p* < 0.05). Conclusions: increasing marginalisation is associated with decreasing health status. Using an intersectional lens can to be a useful way to understand disadvantage for young people belonging to multiple marginalised groups.

## 1. Introduction

Marginalisation is both a process and an experience, consequent upon unequal power relationships where dominant groups within society are privileged over other groups [1]. Those who become “othered” are pushed to the peripheries or the “margins” of mainstream society. This process can involve multiple forms of exclusion, lowering participation in education, work and in healthcare access leading to lower health and social outcomes [1]. Marginalisation is a socio-cultural lens for understanding why some groups experience disadvantage and some groups within society are privileged over other groups [1].

Adolescence is a developmental stage during which biological and cognitive capacity to function autonomously is transformative. However, social and cultural expectations of young people vary enormously, and sanctions which are placed on emerging behaviours can be liberating or oppressive. Adolescence might be a marginalising factor when it comes to health. In the search for identity, which accompanies greater awareness of oneself within a peer group, or local or global community, the social forces which lead to marginalisation (for example due to race, gender or sexual orientation) may be acutely felt by individual young people [2].

Aboriginal young people in Australia make up approximately 5% of the population and have a long history of marginalisation and worse health outcomes [3]. In 2006, 26% of Australia’s young people aged 15–24 lived in inner and outer regional areas, and 2% lived in remote and very remote areas [4]. In 2016, an estimated 25,000 young people aged 12–24 in Australia were homeless, accounting for 24% of the total homeless population [5]. In 2016–2017, Australia welcomed 24,490 humanitarian arrivals, of which 6008 were young people aged 12–25 (25%) [6].

Health issues, such as mental and substance use disorders and chronic physical illnesses, are becoming the dominant health problems of the adolescent age group. Inequalities in health and wellbeing are greater for socially and economically marginalised adolescents [7]. However, for some marginalised groups, there is a lack of health status data available, particularly homeless, refugee and sexuality and/or gender diverse young people [7].

Young people from marginalised backgrounds can have more complex health issues and be vulnerable to ill health due to limited social, educational and economic opportunities [7]. Marginalisation can lead to the underuse of healthcare due to access barriers [8]. Access to health care is a social determinant of health, and many marginalised populations are known to experience barriers that might be due to socioeconomic, social or cultural reasons.

Intersectionality is a conceptual approach to understanding multiple health inequalities in social science and population health research. Arising from black feminist scholarship in the USA, Kimberley Crenshaw, a legal academic, coined the term in 1989 to show how discrimination law failed to recognise individuals who were marginalised in multiple ways [9]. Intersectionality theory recognises that individuals can have multi-dimensional social identities which reflect underlying power structures that produce inequality, and that inequality is thus more than the sum of individual risk factors [10]. This concept sits well within an ecological framework for understanding health and its determinants. 

This study aimed to measure the associations between young people’s health status and belonging to one or more of the following marginalised groups: Aboriginal and/or Torres Strait Islander; living in rural/remote areas; homeless or at-risk of homelessness; refugee or vulnerable migrants; identifying as sexuality and/or gender diverse. The study also aimed to explore intersectionality by measuring the relationship between health status and increased levels of disadvantage.

## 2. Methods

To directly inform new policy in the state of New South Wales (NSW), Australia, this cross-sectional study, the NSW Youth Health Access survey [11], was part of a larger project (“Access 3”) which was designed to explore barriers to access to healthcare, health system navigation, and the role of technology. The methods have been fully described in the study protocol [12], and a summary is provided below.

### 2.1. Recruitment and Completion

The target group was young people (12–24 years) residing in NSW, Australia with oversampling of those who belonged to one of the five marginalised groups listed above. Our target sample size was 350 from each of the five marginalised groups plus 350 who did not belong to any of these groups, aiming for a total sample of 2100. We used a combination of convenience and snowball sampling methods.

Participants were recruited online (targeted emails to relevant networks and via Facebook, Twitter and Instagram) and offline (in youth accommodation services and various youth forums). To purposively sample marginalised young people, we worked with advocates from a range of organisations. Participants went into a draw to win one of 20 vouchers worth AUD 50 (EUR 31).

Between February 2016 and February 2017, 2100 commenced and 1416 participants (12–24 years) completed the survey, with 1012 online and 404 on paper. We achieved our goal of oversampling marginalised groups with 897 (63.3%) participants belonging to at least one marginalised group. We exceeded our target sample size (*n* = 350) for rural/remote (33.9%) and for sexuality and/or gender diverse (30.1%) young people (including sexuality diverse (27.3%), gender diverse (3.0%) and intersex (1.0%)). We fell short of our target sample size for the other three groups: Aboriginal/Torres Strait Islander (12.0%), homeless (8.4%) and refugee (5.3%). The median age was 18 years (IQR 16 to 20). Participants’ gender included female (68.3%), male (28.7%) and other genders (3.0%).

Participants belonging to the five target subpopulations of marginalised groups were identified as follows. Aboriginal and/ or Torres Strait Islander identity was by survey response to the options of Aboriginal, Torres Strait Islander, neither, both and not sure. Rural and remote status was determined using self-reported postcode data and the Australian Bureau of Statistics Standard Geographic Classification (ASGC) Remoteness Structure, including rural (inner and outer regional) and remote (remote and very remote) [13]. Homelessness was identified based on reponse to questions on living situation (e.g., living with relatives, friends, in foster care, in a refuge/supported accommodation, boarding house, on the street/outside), using the Australian cultural definition of homelessness, which contends that homelessness must be understood in relation to housing conventions of a particular culture [14]. Refugee background was distinguished by asking directly about entering Australia as a refugee or asylum seeker, and if ”unsure” country of birth was used additionally to imply vulnerable migrant status. Sexuality diverse participants were identified by questions about sexual identity and attraction. Gender diversity was based on a question asking participants to identify if they were female, male or other (with the capacity for free text answers). Intersex variation was included as a separate question.

### 2.2. Health Status Measures

Health status was measured using the following instruments and questions. Participants were asked to rate their health (“excellent”, “very good”, “good”, “fair” or “poor”) and to report select chronic mental and physical health conditions from a predefined list.

The Kessler-10 (K10) questionnaire [15,16] is a validated instrument with robust psychometric properties that provides a measure of non-specific psychological distress in adolescents [17] and adults, relating to symptoms of anxiety and depression experienced in the most recent 4-week period. The 10-item questionnaire is scored, with total score being classified into “low” (10–15), “moderate” (16–21), “high” (22–29) and “very high” (30–50) levels of psychological distress.

The World Health Organisation Wellbeing Index (WHO-5) is a 5-item validated questionnaire with good construct and predictive validity and high diagnostic accuracy that measures current wellbeing and is validated as a screening tool for depression, with normal mood defined as a score of 51–100, low mood as 29–50, and likely depression as 0–28 [18]. We also used time away from school or work due to illness or injury’ as a proxy for poor health status.

### 2.3. Analyses

There is debate about which statistical methods are best suited to analysing and interpreting quantitative data related to intersectionality [19]. In this analysis, we have taken the simple approach of treating the number of marginalised groups (as defined above) that an individual belonged to as an independent variable. Health status measures were dependent variables. We used SPSS version 24 [20], for all analyses and alpha was set at 0.05. To examine the relationship between health status and marginalisation, we conducted chi-square analyses.

### 2.4. Ethics

Ethics approval was obtained from both the University of Sydney Human Research Ethics Committee (2015/874) and NSW Aboriginal Health and Medical Research Council Ethics Committee (1142/15). The ethical considerations of recruitment and engagement of marginalized young people are described in detail elsewhere [21].

## 3. Results

### 3.1. Demographics

The sociodemographic characteristics of sample have previously been published [11].

### 3.2. Intersectionality

A total of 897 participants (63.3%) belonged to one or more of the five marginalised groups of interest; 519 (36.7%) did not belong to any of the identified marginalised groups. Of those 63.3% belonging to at least one marginalised group, 574 (40.5%) belonged to one group; 281 (19.8%) belonged to two groups and 42 (3.0%) belonged to three or four groups. There was a significant interaction between intersectionality and age (*p* < 0.001). Specifically, the proportion of younger participants (12 to 17 years) increased linearly from 43% among those not belonging to a marginalised group, up to 76% of those belonging to three or four groups (*p* < 0.001). Table 1 describes the intersections between marginalised groups.

### 3.3. Health Status

Health status measures for the whole sample are reported in Table 2. Our survey found 80.8% of participants rated their health as “excellent”, “very good” or “good”. Marginalised participants were less likely to rate their health as “excellent”, “very good” or “good” (79.6% vs. non-marginalised 82.8%, *p* = 0.035). See Table 2.

#### 3.3.1. Chronic Health Conditions

Over 50% of participants reported at least one chronic health condition or disability. Marginalised participants were more likely to have a chronic health condition (55.4%, 497/897) compared with the non-marginalised group (46.1%, 239/519, *p* = 0.001). Among those who had a chronic health condition or disability (*n* = 736), 63.2% had one chronic health condition or disability, 27.7% had two, 6.9% had three and 2.2% had four or more. The number of health conditions increased with increased marginalisation (*p* = 0.001). Substance use was associated most strongly with increasing marginalisation: 2.3% for no marginalised groups, 3.7% for 1 group, 6.0% for 2 groups and 23.8% for participants who belonged to 3 or 4 groups (*p* < 0.001) (see Table 2).

#### 3.3.2. Psychological Distress

Among the entire sample (*n* = 1416), 1400 participants completed the K10 questionnaire. A total of 729 (52.1%) had high/very high K10 scores. Those who identified with at least one marginalised group were significantly more likely to have high or very high K10 scores than those who did not belong to any of the five marginalised groups (57.1% vs. 43.4%, *p* < 0.001). The rates of very high levels of psychological distress measured by the K10 also increased with belonging to an increasing number of marginalised groups: 22.6% for no marginalised groups, 31.9% for 1 group, 33.0% for 2 groups and 46.2% for participants who belonged to 3 or 4 groups (*p* < 0.001; see Table 2).

#### 3.3.3. Wellbeing

The World Health Organisation Wellbeing Index (WHO-5) was completed by 1403 participants. There was no association between WHO-5 score and belonging to a marginalised group (likely depression: belong to marginalised group 20.2% vs. does not belong to marginalised group 17.4%; *p* = 0.324). There was no association between belonging to an increasing number of marginalised groups and WHO-5 wellbeing score. Wellbeing was significantly lower among participants who had a chronic condition and/or disability (28.2% vs. 17.3%, *p* < 0.001(see Table 2).

#### 3.3.4. Time Away from School or Work Due to Illness or Injury

A total of 604 participants (42.7%) had stayed away from school or work in the month prior to survey due to illness or injury. Marginalised participants did not significantly differ from non-marginalised participants in terms of the proportion with time away from school or work (marginalised: 44.5%, *n* = 398/895; non-marginalised: 39.7%, 206/519; *p* = 0.08). Spending time away from school or work due to illness or injury increased with increasing marginalisation (*p* = 0.032).

### 3.4. Gender, Marginalisation and Health Status

Associations between gender, intersectionality and health status were explored. There were more females (*n* = 968) than males (*n* = 406), and an age and gender interaction across the sample, with females being older and males younger. The “other” gender category was a much smaller group (*n* = 42) and by definition were a subset of the sexuality and gender diverse group, which also confounds analysis. However, when adding/ controlling for gender, the interactions between self-reported health status, self-reported chronic health condition or disability, psychological distress and the WHO-5 wellbeing index, and time away from school or work due to illness or injury, the association with marginalisation only held for females, not males.

## 4. Discussion

This is the first Australian study of young people’s self-reported health status which has included substantial numbers of young people identified with at least one marginalised group. The sample allowed for an exploration of intersectionality on health status, including the compounding effect of belong to multiple marginalised groups.

Almost two-thirds of participants (63.3%) belonged to one or more of the five marginalised groups of interest with (19.8%) belonging to two groups and (3.0%) belonging to three or four groups.

While 80.8% of participants rated their health as “excellent”, “very good” or “good”, this was lower than previous population surveys: in 2014–2015, where 93% of Australian young people aged 15–24 years rated their health as “excellent”, “very good” or “good” [22]. Participants also had significantly higher levels of psychological distress with 52.1% having high or very high K10 scores, when compared to 11.7% of young Australians in the Australian census data [22]. The comparatively poor health status of participants can be explained by the sampling method (self-selection) and oversampling of marginalised groups, given the levels of high or very high mental distress and chronic health conditions in these groups. These findings are consistent with other studies of young people which have observed less favourable mental health within marginalised groups, including those who are Aboriginal and Torres Strait Islander [23], living in rural and remote areas [24], homeless [25], and sexuality and gender diverse [26]. Another study with adult refugees, aged 18 and above living in Australia, also found indicators of poorer mental health [27].

Poorer self-reported health status, the number of self-reported health conditions, and high or very high rates of psychological distress were associated with an increasing number of marginalised groups. Intersectionality theory may be used to explain these results, namely, that belonging to multiple marginalised groups leads to greater inequality, and therefore inequality is more than the sum of individual risk factors [10].

An anomaly in our findings was that multiple marginalisation was not associated with WHO-5 wellbeing scores. This was unexpected given that the WHO-5 is validated as a tool to identify depression and anxiety [28]. This finding could be an artefact, or may warrant further investigation into psychometric properties of screening tests for mental health problems in populations who are marginalised in multiple ways. However, while there was no association between belonging to an increasing number of marginalised groups and WHO-5 wellbeing score, the results for WHO-5 scores correlated with having a chronic condition/disability. This is consistent with previous studies which have found a link between WHO-5 scores for young people with diabetes [29] or a disability (especially for young people with multiple or severe impairments) [30].

Although intersectionality has been described conceptually, approaches to analysis in quantitative studies of health outcomes are still developing. Seng et al. [31] have proposed that intersectionality can be conceptually and statistically linked across socio-ecological levels. For example, marginalised status may relate to the levels described in Bronfenbrenner’s socio-ecological model: intrapersonal; interpersonal; contextual; and structural.

Multiple adverse events in childhood (ACEs) are associated with poorer long-term health outcomes, with a direct relationship between the number of adverse experiences and likelihood of inferior health outcomes [32,33] Many interventions to improve outcomes for people who have experienced ACEs focus on mitigating individual psychological harms [34]. Although adverse experiences in childhood and belonging to marginalised groups are clearly separate concepts, it is likely that adverse experiences occur as a result of being marginalised. Belonging to multiple marginalised groups may therefore make adverse experiences even more likely.

In a linked Access 3 study we found that increasing marginalisation was inversely associated with reporting of the number of healthcare access barriers [11]. This could be a good news story that highly marginalised young people are reaching the healthcare required. However, health outcomes, as measured by self-reported health status, are not showing the dividends that might be expected from having fewer healthcare access barriers. Ill health, may therefore, be linked with broader factors associated with wellbeing from a socioecological viewpoint [35], such as stable housing location, economic and educational participation, social inclusion and community connection.

### 4.1. Implications for Policy, Practice and Research

The Access 3 study has policy and practice implications relevant to marginalised young people and particularly those belonging to multiple marginalised groups. Health resources are limited, posing a challenge for health policymakers in the allocation of resources. It is important to understand who are most at-risk of poorer health outcomes. An intersectional perspective was found to be a useful way to understand disadvantage for young people belonging to multiple marginalised groups and to appreciate that social determinants of health intersect and compound poor health status.

Policies that aim to reduce disadvantage, for example homelessness, need to take an intersectional approach and address stigma and discrimination associated with young people who belong to other marginalised groups such as being sexuality and/or gender diverse, Aboriginal, or refugee. An increase in appropriately funded targeted resources for marginalised young people in urban and rural areas would also be of benefit. The silo-ed nature of funding and therefore service delivery in the Australian health system might also present barriers to addressing intersectionality in policy and practice. Frameworks and practice guidelines encouraging integration and addressing multiple disadvantages would be of benefit [36].

As health systems recognise and respond to the specific needs of young people, it is important to ensure equity of access for all groups. In our systematic review [37] we found that studies into access and health system navigation for marginalised young people almost always focused on only one group, yet it is likely that multiple marginalisation is not an uncommon experience for young people.

Health care providers may not always have the knowledge or tools to address these social identities. In a related qualitative Access 3 study with health professionals, the possibility and impact of young people belonging to multiple marginalised groups was largely overlooked [38]. An intersectional approach may lead to recognition of the additional needs of marginalised young people and, together with the provision of non-judgmental and respectful services, professionals will be better able to respond to the health care needs of the marginalised and vulnerable.

A shift in focus from individual harms—to disadvantage experienced by groups—may also shift the focus for the level of interventions. While addressing psychological harms commonly focusses on individual therapies [34], interventions for reducing social disadvantage and/or addressing its effects are more likely to involve public health approaches. For example, public health interventions to improve outcomes for gender and /or sexuality diverse young people may involve addressing homophobia and transphobia in communities, rather than providing individuals with psychological support [39].

Appropriate attention by health professionals, policymakers, and researchers would be helpful to more adequately support the needs of marginalised young people. In a systematic review of marginalised young people’s healthcare access, “other” gender status was frequently not collected [11]. Thus, when considering health inequalities and gender, if gender is conceptualized only as a binary concept, this vulnerable group will be overlooked [40]. A very useful first step would be awareness and inclusiveness in assessment, collection and analysis of data on “other” genders to better understand the specific needs of this group. Access to appropriate specialty services may also reduce stress and vulnerability.

### 4.2. Limitations

This was a self-report and cross-sectional survey, with both methodologies having known limitations. Even as a convenience sample, we used different recruitment strategies which led to selective samples. We oversampled marginalised groups, which likely reflects the poorer health status and health ratings than other Australian population health surveys. This finding may place some limits on generalizability, including that the number of those who belonged to three or more marginalised conditions was small.

## 5. Conclusions

Increasing marginalisation, as defined by belonging to a higher number of marginalised groups, is associated with poorer health status. An intersectional lens explores how individuals can have multiple social identities which impact on health equalities. Typically research into associations between young people’s health needs and care access has only explored individual marginalised groups. An intersectionality approach, which considers the impact of multiple marginalisation, may help in identifying those who are most at-risk of poorer health outcomes. Researchers can further explore the best methods to identify these social identities, and investigate ways to use the intersectionality literature to interpret health data according to increasing disadvantage.

## Figures and Tables

**Table 1 ijerph-17-08104-t001:** Intersections between marginalised groups.

	Rural		Sexuality and/or Gender Diverse		Aboriginal and/or Torres Strait Islander		Homeless		Refugee	
	Yes *n* = 478	No *n* = 930	Yes *n* = 42	No *n* = 988	Yes *n* = 167	No *n* = 1240	Yes *n* = 118	No *n* = 1290	Yes *n* = 75	No *n* = 1333
Aboriginal and/or Torres Strait Islander	142 (29.8%)	25 (2.7%) ‡	25 (5.9%)	143 (14.5%) ‡	N/A	N/A	25 (21.2%)	141 (10.9%) ‡	N/A	N/A
Rural	N/A	N/A	118 (27.8%)	358 (36.5%) ‡	142 (85.0%)	335 (27.0%) ‡	39 (33.9%)	435 (33.8%)	7 (9.5%)	471 (35.4%) ‡
Homeless	39 (8.2%)	76 (8.2%)	37 (8.7%)	81 (8.2%)	25 (15.1%)	93 (7.5%) ‡	N/A	N/A	8 (10.7%)	106 (8.0%)
Refugee	7 (1.5%)	67 (7.2%) ‡	16 (3.8%)	59 (6.0%)	N/A	N/A	8 (7.0%)	67 (5.2%)	N/A	N/A
Sexuality and/or gender diverse	118 (24.8%)	306 (32.9%) ‡	N/A	N/A	25 (14.9%)	400 (32.2%) ‡	37 (31.4%)	387 (30.0%)	16 (21.3%)	406 (30.5%)

‡ *p* < 0.01; N/A, not applicable.

**Table 2 ijerph-17-08104-t002:** Health status by number of marginalised groups.

	Did Not Belong to Any of the Identified Marginalised Groups *n* (%)	Belonged to One Marginalised Group *n* (%)	Belonged to Two Marginalised Groups *n* (%)	Belonged to Three or More Marginalised Groups *n* (%)	Total *n* (%)
**Self-reported health rating (N = 1410)**					
Poor	18 (3.5)	28 (4.9)	10 (3.6)	6 (14.3)	62 (4.4)
Fair	71 (13.7)	71 (12.4)	59 (21.1)	8 (19.0)	209 (14.8)
Good	176 (34.0)	223 (39.1)	111 (39.8)	11 (26.2)	521 (37.0)
Very good	195 (37.6)	193 (33.8)	63 (22.6)	10 (23.8)	461 (32.7)
Excellent	58 (11.2)	56 (9.8)	36 (12.9)	7 (16.7)	157 (11.1)
**Chronic health conditions (self-reported) (N = 1416)**					
No chronic health conditions	309 (59.5)	277 (48.3)	142 (50.5)	21 (50.0)	749 (52.9)
Mental health condition only	125 (24.1)	196 (34.1)	75 (26.7)	13 (31.0)	409 (28.9)
Physical health condition only	55 (10.6)	54 (9.4)	25 (8.9)	2 (4.8)	136 (9.6)
Both mental and physical health condition	30 (5.8)	47 (8.2)	39 (13.9)	6 (14.3)	122 (8.6)
**Level of psychological distress (K10 score) (N = 1400)**					
Low (10–15)	158 (30.5)	122 (21.5)	63 (22.8)	11 (28.2)	354 (25.3)
Moderate (16–21)	135 (26.1)	121 (21.3)	57 (20.7)	4 (10.3)	317 (22.6)
High (22–29)	108 (20.8)	143 (25.2)	65 (23.6)	6 (15.4)	322 (23.0)
Very high (30–50)	117 (22.6)	181 (31.9)	91 (33.0)	18 (46.2)	407 (29.1)
**Wellbeing (WHO-5 score) (*n* = 1403)**					
likely depression	90 (17.4)	110 (19.3)	60 (21.7)	9 (23.1)	269 (19.2)
low mood	121 (23.4)	146 (25.7)	63 (22.7)	5 (12.8)	335 (23.9)
not likely to have depression	307 (59.3)	313 (55.0)	154 (55.6)	25 (64.1)	799 (56.9)

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
