# Peer review of "Intersectionality: Social Marginalisation and Self-Reported Health Status in Young People"

_ijerph, 2020, doi:10.3390/ijerph17218104_

Round 1
Reviewer 1 Report
I think that the introduction is poor. The introduction is too brief especially in relation to the discussion. This article belongs to a larger study, so the introduction may not be proportional to the discussion.
Data are lacking to complete the consistency of the study.
Thank you.
Kind regards.
Reviewer 2 Report
Thank you for submitting your manuscript to the International Journal of Environmental Research and Public Health. Below are my comments:
Thank you for submitting your manuscript to the International Journal of Environmental Research and Public Health. Below are my comments:
- Please pay attention to punctuation throughout the manuscript.
- Please edit the abstract for clarity and flow. Write New South Wales instead of NSW.
- Please add more references in the introduction section.
- Please add some statistics on the population studied, especially in NSW. Also add why these populations [Aboriginal and/or Torres Strait Islander; living in rural/remote areas; homeless or at-risk of homelessness; refugee or vulnerable 61migrants; identifying as sexuality and/or gender diverse] are considered as marginalized.
- Methods: Please divide the scales used into separate sections. For example, demographics, socio-economic status, employment, health, mental health etc.
- Please add reliability for each scale used in the study. It is not enough to state that the alpha was set at 0.05.
- Ethical considerations need to be added to the manuscript. It is not enough that the study was approved by the research committee. Authors need to articulate any usage of consent form, language of the survey and time for survey completion. In addition, due to the sensitive population studied, what were the authors’ procedures and conduct to making sure that no harm was caused to participants due to their exposure to the study’s questions. Also, what did the authors do when participants scored high on mental health and health measures? Were they referred to any treating organizations? Any follow ups?
- Recruitment and Completion section needs to be part of the methods section, not the results.
- Results section needs to start with basic demographics. It is better to present these in a table. The range between 12 and 24 years of age is too large. I would recommend to divide it at least to two age rage [12-17] and [18-24] since these ages may differ enormously in regards to health and mental health outcomes. If this is too complex to do, then at least run the analysis to check for differences and declare them.
- Please provide the limitations of the study as a separate section.
- The analysis remains in the descriptive level, including all tables provided. I would expect from such a study with this large sample size for a more complex analysis that can better clarify the differences among variables. Such analysis will better serve these populations in providing some recommendation to policy makers and care providers.
- Lines 223-229 – please provide references to support your claims!
- The conclusions and part of the discussion are not based upon the study’s findings!!!
Reviewer 3 Report
Abstract:
Page 1, line 17, suggest replacing the word "quantify".
Page 1, line 20, suggest to spell out the abbreviation "NSW".
Page 1, line 29, suggests rephrasing the sentence and changing the "poorer" word.
Introduction: (i) The paragraphs seem disconnected – suggest adding transitional sentences for better flows. (ii) The words "poorer" and "quantify" do not seem to be the right choice – recommend replacing them. (iii) The authors did perform and report the intersection (or correlation, as my understanding) among the marginalized groups or independent variables. Perhaps, this analysis needs to be included in the introduction section.
Page 1, lines 39 – 41, if adolescents and young adults are different age groups, suggest adding a definition of each term, e.g., age range.
Page 1, line 42, suggest deleting the word "itself".
Page 2, lines 46 – 48, suggest shortening the sentence by cut it into 2.
Page 2, line 59, suggest replacing the work "quantify".
Methods: As mentioned in the introduction section's suggestion, the author may add the intersection analysis among the marginalized groups (independent variables).
Page 2, line 64, the authors need to spell out the abbreviation NSW first time it is introduced.
Page 2, line 79, the sentence need rephrasing. Perhaps… "was operationalized by survey…"
Results: (i) Table 2 needs footnote to explain what numbers within the parentheses present. (the response percentage within each independent variable?). (ii) There is a need for a table to demonstrate the chi-squared analysis between independent and dependent variables, such as c2 measures and p-values. (iii) The word "sample" was used interchangeably with "participants", suggest replacing it with "participants" throughout the article.
Page 3, lines 123 – 126, suggest to cut the sentence into three and rearrange the components.
Discussion:
Page 6 – 7, lines 223 – 236 and 243 – 253, I do not see these paragraphs' relevance within the study's content.
Page 7, lines 237 – 242, this paragraph is essentials; however, it did seem to be out of place.
Conclusion: The mentioning of "intersection" reiterates the need to add this analysis's description in the methods section.
Round 2
Reviewer 3 Report
Thank you!